# How can we strengthen mental health services in Swedish youth clinics? A health policy and systems study protocol

Linda Richter Sundberg [1], Monica Christianson,[2] Maria Wiklund,[3] Anna-Karin Hurtig,[1] Isabel Goicolea[1]

¹Department of Epidemiology and Global Health, Umeå University, Umeå, Sweden
²Department of Nursing, Umeå University, Umeå, Sweden
³Department of Community Medicine and Rehabilitation, unit of Physiotherapy, Umeå University, Umeå, Sweden

**Correspondence to**
Dr Linda Richter Sundberg;
linda.sundberg@umu.se

## ABSTRACT

**Introduction** Strengthening first-line mental healthcare services for youth remains a priority for the Swedish government. The government is currently investigating how different sectors involved can be strengthened, but evidence is scarce. Youth clinics play a key role in these discussions, being one of the most trusted services for youth. However, analysis of organisational functions and coordination with other services is important to strengthen youth clinics' role in first-line mental healthcare. This study investigates these challenges and aims to analyse the integration of mental healthcare within youth clinics to identify strategies to strengthen first-line mental healthcare for youth in Sweden.

**Methods and analysis** This study adopts a health policy and systems approach. In the first phase, a formative realist evaluation is conducted to ascertain what works in terms of integrating mental healthcare services within youth clinics, for what type of youth subpopulations and under what circumstances. National-level stakeholders will be interviewed to elicit the programme theory that explains how the intervention is supposed to work. The programme theory will then be tested in three–five cases. The cases will be comprised of youth clinics and their stakeholders. Quantitative and qualitative information will be gathered, including via visual methodologies and questionnaires. The second phase includes a concept mapping study, engaging stakeholders and young people to build consensus on strategies to strengthen the integration of mental healthcare into youth clinics.

**Ethics and dissemination** The Swedish Ethical Review Authority has approved the study (2019-02910 and 2020-04720). The results will be published in open-access peer-reviewed journals and presented at scientific conferences.

## INTRODUCTION

During the last decade, the increase of mental health problems among youth is evident in different parts of the world.[1 2] In parallel, there are signs of health systems struggling to adequately respond to youth mental health needs, in particular when implementing early interventions.[3 4] More than 75% of mental health problems have onset during childhood or adolescence; therefore, this is a period of opportunity for health systems to promote youth mental health and respond to

---

### Strengths and limitations of this study

► The study addresses an urgent problem in health systems globally—the barriers to youth accessing high-quality mental health services.
► The study will generate knowledge to strengthen first-line mental healthcare for youth in Sweden and in similar settings where differentiated services for youth that integrate mental healthcare are being developed.
► This study will add the team and organisational levels to the individual level, which might challenge the method of realist evaluation to some degree.
► A challenge of the study is to ensure meaningful participation of young people, especially in the concept mapping step of the study.

---

youth's mental health problems in coordination with other actors and sectors.[5 6] International research shows that access to mental health services for youths is often hindered by factors such as long waiting times, youth concerns about confidentiality and lack of trust in professionals.[6 7] Young people experience the transition between mental health services (eg, from child to adult mental health services) as poorly coordinated and dramatic.[8 9]

In Sweden, the prevalence of mental health problems among young people has increased dramatically by about 70% during the period 2006–2016, the highest increase among the Nordic countries. Depression, anxiety and substance abuse have shown the largest increase,[10 11] and suicide rates have not decreased in younger age groups unlike in the overall population.[12]

To enhance accessibility to mental health services for youth and facilitate coordination, the Swedish health system since 2009 has been developing a strategy to integrate youth mental health services within primary care services. The integration is labelled first-line mental health services (FLMHS).[13 14] This

approach has been used in several other countries.[15 16] Research has shown that the integration of mental health services within primary care improves the management of common mental health problems among adults[16 17] and children.[18] Youth between 15 years and 25 years of age have found it easier to access mental health services when delivered through less stigmatised, more visible and familiar primary care services.[19] Reforms of youth mental health services have been implemented in many countries, but surprisingly, there is still limited research analysing integrated care models such as the FLMHS.[19–21]

The FLMHS was launched as a way to address three main challenges: (1) inequities in the mental health and social services provided to children and youth, (2) unclear division of responsibilities within and between health and social services and (3) a need to develop versatile health system solutions reflecting the diverse conditions prevailing in the different groups of the population and parts of the country. The establishment of FLMHS was in addition expected to relieve the high (unfulfilled) demand for specialised child mental health services and strengthen the collaboration between diverse public services and sectors.[22 23] Currently, the sectors that are most frequently involved in FLMHS are primary health-care centres, school health services and youth clinics (YCs).[24] More than 250 YCs have been in operation in Sweden since 1970–1980s, providing youth-centred health services for young people. According to the Swedish Society for Youth Centres (FSUM) guidelines, the minimum staff of a YC includes a midwife and a social counsellor or psychologist, with a physician working part-time. However, many YCs (especially the ones located in bigger cities) also have nurses, dieticians, sexologists and specialised medical doctors (eg, psychiatrists and obstetricians). There can be cooperation with other services such as the school health services and child/adult psychiatry, or the local employment and health insurance offices. Young people contact the YC directly as outpatient visitors without referral, for cost-free consultation. Most YCs are placed off the premises of general health services, and in addition to consultations provided in the clinic they also work in promotion, mainly through school visits of 15-year-old pupils to the clinics.[25 26] This national network of YCs offers an important opportunity to integrate mental health services within services that report a high level of trust by young people.[27] Youth commonly approach YCs with needs and questions that fall within the area of sexual and reproductive health, for example, requests for prescriptions for birth control pills, tests or treatment of sexually transmitted infections. The identity and role of YCs is well established among youth, YC health professionals and in other parts of the health system. However, ever since their establishment in the 1970s, YCs are also responding to youth needs in regard to mental health and mental health problems. YCs have come to play a key role in the FLMHS, where youth mental health services are integrated in primary care level. YCs operate together with other stakeholders within the FLMHS and interact with stakeholders from the larger youth mental health system, for example, authorities, specialised health services and schools (figure 1).

In this paper, we present the protocol for a health policy and system study with an aim to analyse Swedish policy efforts to strengthen FLMHS for youth. We focus on YCs and adopt Atun et al's[28] conceptual framework for investigating the integration of health interventions into health systems. The research questions we aim to answer with this study are as follows: (1) 'how and under what circumstances are YCs part of the FLMHS for youth in the Swedish health system?' and (2) 'how can the integration of mental health services into YCs in the Swedish health system be strengthened?'.

## Conceptual framework

The approach to mental health systems used in this study is inspired by complexity theory,[29] with a specific focus

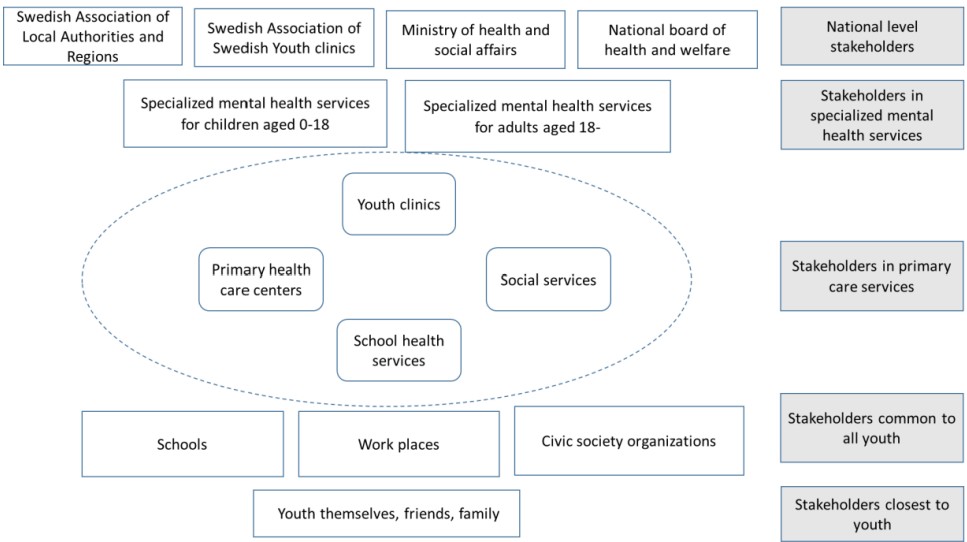

**Figure 1** Stakeholders in and surrounding the first-line mental health services for youth.

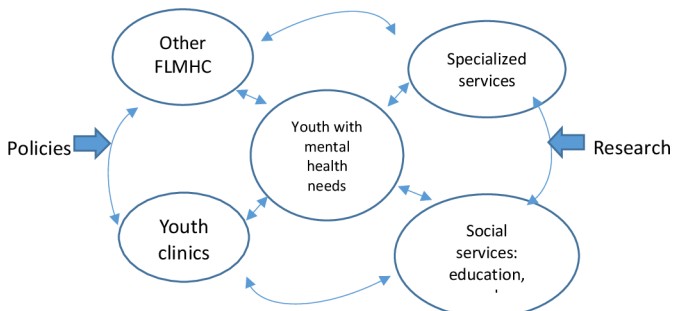

**Figure 2** The model for mental healthcare delivery as a complex adaptive system inspired by Ellis *et al*.[29] FLMHS, first-line mental health services.

on the integration of targeted health interventions within health systems.[28] Instead of portraying the different hierarchical levels and actors in the mental health system,[22] complexity theory portrays them as interconnected and centred on the service user (figure 2). In complex adaptive systems, individuals, services and organisations are active agents that interact with each other, establishing dynamic interconnections.[29]

This study will explore the intentions and strategies among national stakeholders and policy-makers (eg, authorities and politicians) in terms of integrating YCs in FLMHS and if and how YCs coordinate with other health actors in the FLMHS for youth, with specialised mental health services and with social services. The organisation of this system is influenced by, and influences, both policies and research (figure 2).

Inspired by Atun *et al*,[28] we conceive integration as the process, pattern and extent of adoption and eventual assimilation of new strategies into the health system (in this case, mental health services into YCs) that are embedded within FLMHS for youth.

In figure 3, we have adapted Atun *et al*'s[28] framework to the specific intervention being analysed in this study. Atun *et al*'s[28] conceptual framework provides a systematic approach to researching integration by focusing on five domains that influence the adoption and diffusion of health interventions.

The first domain relates to the nature of the problem being addressed, that is, the understanding regarding the urgency and severity of the problem influences the speed of adoption.[28] In this protocol, we conceptualise the problem as the weaknesses of the health system to adequately respond to the increasing mental health needs of young people.

The second domain refers to the characteristics of the intervention. Here, Atun *et al*[28] point out that interventions that, for example, are perceived as having a relative advantage are less complex and risky and are more easily integrated. In this protocol, the intervention to address this problem is the integration of mental health services within the existing network of Swedish YCs.

The third domain refers to the adoption system, namely, the key actors and institutions involved in the integration.[28] Each stakeholder has differing perceptions

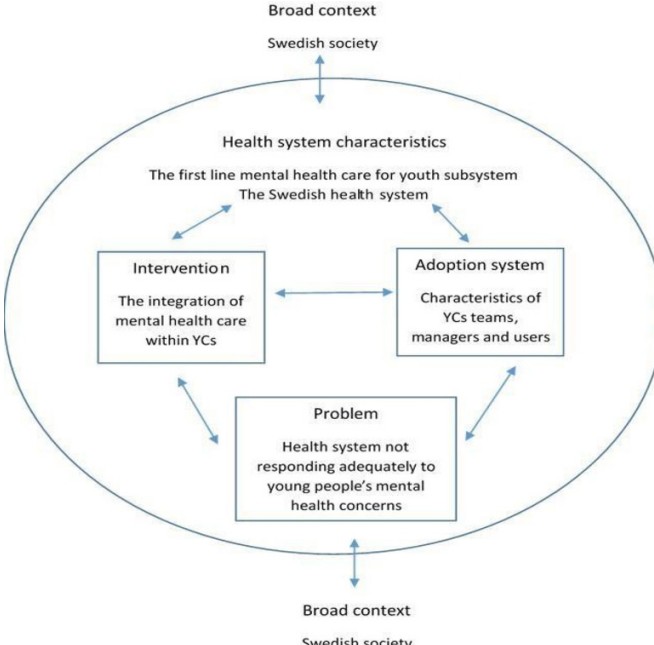

**Figure 3** Conceptual framework to analyse integration of mental healthcare for youth within Swedish YCs. YCs, youth clinics.

of the benefits, risks and legitimacy of the intervention, and the different interests at stake have to be explored. The adoption system in this study focuses on the YC care providers, the managers and the young service users.

The fourth domain refers to the characteristics of the health system.[28] With regard to the characteristics of the health system, we are interested in the larger mental health system (ie, national governance and primary and specialised mental health and social services), but with special focus on the subsystem of FLMHS for youth. The focus is on analysing the collaboration between the services responsible for FLMHS for youth and how the different ways of organising them and their approaches toward young people influence both the integration of mental health services within YCs and the collaborative dynamics with the other services.

The fifth domain refers to the broader sociocultural context, where critical events, prevailing political economy and sociocultural norms also affect integration.[28] The broader context in the study protocol refers to how the current political economic situation and the sociocultural characteristics of Swedish society influence the integration of mental health services for youth within YCs.

## MATERIAL AND METHODS
### Study design
To address our two main research questions, two methodological approaches will be combined in a sequential way: (1) a realist evaluation[30] and (2) concept mapping.[31] The study design will follow an iterative analytical process in which the conduct of the concept mapping will draw on

**Table 1** The research process

| Research question in focus | Methodological approach | Steps | Methods for data collection | Methods for data analysis |
|---|---|---|---|---|
| How and under what circumstances are YCs part of the FLMHS for youth in the Swedish healthcare system? | Realist evaluation | Eliciting the programme theory | Literature review<br>Documentary review<br>Interviews with stakeholders | Thematic analysis |
| | | Testing the programme theory in selected cases | Analytical multiple case study<br>Questionnaires for mental healthcare consultations | Descriptive statistical analysis |
| | | Refining the programme theory | Interviews with professionals from YCs<br>Interviews with professionals from other services involved in FLMHS for youth<br>Interviews with professionals at specialised services<br>Photo elicitation with young people | Thematic analysis<br>Thematic analysis |
| How can the integration of mental healthcare services into YCs in the Swedish health system be strengthened? | Concept mapping | Presenting the findings and engaging professionals and young people | Workshop and photo exhibition with (1) professionals working in YCs, (2) professionals in other FLMHS services for youth and in managerial positions and (3) young people and youth organisations | |
| | | Developing the concept maps | Brainstorming, pile sorting and rating, and map development using Concept System-CS Global Max | Concept mapping techniques (mixed methodology combining multivariate statistical methods and collective qualitative interpretation) |
| | | Bringing everything together and proposing actions | Final workshop | |

FLMHS, first-line mental health services; YCs, youth clinics.

the findings from the realist evaluation. Qualitative and quantitative data collection and analysis will be combined to best capture the complexity of the phenomenon under investigation. The integration of qualitative and quantitative approaches enables researchers to explore phenomena from different perspectives and provides an enriched understanding.[32] The data collection started in November 2019 and will be finalised in November 2022. The main research questions and the methodologies applied are summarised in table 1.

### Patient and public involvement
The research questions in this study have been informed by the experiences and preferences of youth, through our previous studies exploring youth views on YCs. Furthermore, the study adopts a participatory approach, which has implications for the design, conduct of the study and also the choice of outcome measures.

All phases of the study will be presented and discussed with stakeholders at YCs (both managers and professionals) and other arenas (ie, youth organisations) before and during data collection. Dissemination is a core component of the second phase. Stakeholders (including young people) will be invited to workshops to discuss preliminary results and engage in a concept mapping to collectively develop a set of strategies to improve the integration of mental health services within YCs. One of the main outcomes in this study is to reach consensus on

the strategies to strengthen the integration of YCs in the FLMHS.

### Study setting
The responsibility for providing health services in the Swedish healthcare system is shared between the national government, the regions and the municipalities. The Swedish Healthcare Act gives a commitment to provide 'good health and care on equal terms for the entire population'.[33] However, evaluations have shown that evidence-based health services are inaccessible to children and youth with mental health problems to an extent that diverges from the Healthcare Act.[34]

### Data collection and analysis phase I: formative realist evaluation
To answer the first research question, a formative realist evaluation of the integration of mental health services within YCs will be conducted. Realist evaluation is well suited to analysis of complex interventions aiming to ascertain whether the intervention works but also why, for whom and under what circumstances.[30] In realist evaluation, theories are at the middle-range level, connecting the intervention with contextual factors, with the mechanisms that the intervention triggers within the target populations and with the outcomes achieved. The aim is to identify patterns of intervention–context–mechanism–outcomes

that can explain how, why and under what circumstances a specific intervention works.

Pawson[30] describes cycles of realist evaluation, starting from eliciting the programme theory behind the implementation of a particular intervention, followed by putting this theory under test in specific cases and coming up with a refined programme theory. The entire process should be conducted in close dialogue with stakeholders involved in the development and implementation of the intervention, because the aim is to come up with a revised and improved intervention.[30]

Realist evaluation can be applied in a summative way (once an intervention has ended or come to a conclusion) or in a formative way (while an intervention is being implemented). Integrated care programmes for young people's mental health need to adapt and learn continuously; therefore, we view our realist evaluation through the lens of formative evaluations, aiming for continuous quality improvement. The realist evaluation will cover the following steps: (1) developing the programme theory, (2) testing the programme theory in selected cases and (3) refining the programme theory.

### Developing the program theory

The realist evaluation will start developing the programme theory through a review of the literature and policy documents. The scientific literature, policies and other relevant documents related to the intervention will be reviewed, looking for key concepts, ideas and theories that can help to explain how the integration of mental health services for youth within the FLMHS is supposed to work. The scientific literature and documents will be analysed using thematic analysis following Clarke and Braun.[35]

In addition, we will interview national-level and regional-level stakeholders involved in developing the intervention (ie, the Swedish Association of Local Authorities and Regions, and the Swedish Association of Youth Clinics). The interviews will aim to make the programme theory behind the intervention explicit, namely, the integration of mental health services within YCs, and to analyse how the increasing problems of youth mental health and the planned solutions/responses are understood. The five domains of Atun *et al*'s[28] framework in relation to the three key concepts of realist evaluation (context, mechanisms and outcomes) will be the main topics explored. The interviews will be audio-recoded, transcribed and analysed using abductive thematic analysis,[35] moving back and forward between our conceptual framework and the data. The programme theory that develops will explain how the intervention is supposed to work. Following the ideas of FLMHS,[22] the outcomes will mainly focus on whether the YCs (1) are accessible, (2) allow early detection, (3) have broad expertise of good quality, (4) have good reporting and monitoring systems and (5) have good collaboration with other services.[22]

### Testing the programme theory in selected cases

In the next phase, the programme theory will be put to the test in three–five specific cases: YCs and the FLMHS in which they are embedded, including the surrounding referral services. To capture diverse contextual circumstances, we will choose clinics in different municipalities, located in different regions and embedded in different models of FLMHS for youth (as defined by the Swedish Association of Local Authorities and Regions[22]).

From each case, quantitative information will be gathered on the extent and characteristics of mental health consultations. For this study, we will distribute the questionnaire to all the professionals working at the YC in the three–five cases, including midwives, psychologists, social workers, medical doctors and other staff who are frequently approached by young people to discuss mental health issues. The questionnaire will be applied over 6 months in each of the clinics participating as cases. Analysis of the quantitative data will offer answers whether the services (1) are accessible, (2) allow early detection, and (3) offer good quality services.[23]

Qualitative information from each case will include interviews (~5–10 per case) with (1) professionals working in the YC, (2) professionals working in other FLMHS for youth (eg, student health, health centres and social services) and (3) professionals working in specialised mental health services. Because the way different municipalities are implementing FLMHS for youth and the services with responsibility for this subsystem vary, the institutions represented might vary from case to case. Professionals working in the YCs will include the head of the clinic and a variety of professionals, including professions more focused on mental health (eg, counsellor and psychologist) and others (eg, midwives). Interviews at this stage will focus on health professionals' experiences of mental health consultations and if and how mental health services have been integrated and possibly sustained within the clinic's practice, coordination within the clinic and with other healthcare resources (both first line and specialised) and social resources. Professionals working in FLMHS for youth will include those working in, for example, primary healthcare centres and school health services. Specialist mental health professionals will include those working in child mental health services and adult mental health services. Interviews with staff in specialised mental health services will focus on how they perceive the role of YCs in FLMHS for youth, and their collaboration with YCs and with other services. Data from the interviews will be analysed using thematic analysis.[35]

To capture the views and experiences of young people (aged: 16–29 years), we will engage young people in an auto-driven photo-elicitation interview to gather their perspectives.[36] Photo elicitation is a research method that incorporates images in the interview process to elicit participants' subjective explanations. Photo elicitation has gained recognition as a way to better capture young people's attention, ease rapport and balance power differentials.

From each case, 6–8 youths aged 16–29 years will be recruited. Participants will include youth with experience using YC mental health services and those who do not have that experience. For the latter, we will recruit young people from high schools and universities. First, each participant will be approached and receive information about the study, offered a digital camera for 2 months and asked to take pictures to depict young people experiences and/or expectations of mental health and/or mental health services in the area where they live. At a second meeting, the participant will select three of the pictures taken as the basis for an interview, where topics around youth mental health, access to services and experiences of them (with special focus on YCs) will be discussed.

All the interviews will be audio-recorded and transcribed. Detailed descriptions of the cases will be developed and further analysed using an abductive thematic analysis approach, guided by the framework presented above (figure 2). This analysis will inform the intervention implementation, contextual factors, potential mechanisms and outcomes.

### Refining the program theory

The initial programme theory will be refined based on the findings in the theory testing to specify a middle-range theory that describes what works in terms of integrating mental health services for young people within YCs, for what type of youth subpopulations and under what contextual circumstances.[30 37]

### Data collection and analysis phase II: a concept mapping study

Building on the knowledge gained during the previous phase, we will engage in a participatory process in which professionals and young people will be invited to take part in workshops followed by a concept mapping study.[31] The aim of this study is to develop strategies to strengthen the integration of mental health services within YCs. The concept mapping entails three steps: presenting findings and engaging professionals and young people in participatory workshops, developing the concept maps and bringing everything together and proposing actions.

### Presenting findings and engaging professionals and young people in participatory workshops

To present the preliminary results of the study, workshops will be arranged with three different types of stakeholders: (1) professionals working in YCs, (2) professionals in other FLMHS for youth and in managerial positions and (3) young people and youth organisations. Youth will be recruited to the workshop using three strategies: first, the youth who were involved in the previous interview study in the project will be asked to participate in the workshop; second, YCs involved in the case studies will be asked to invite youth through advertising in waiting rooms/reception areas; and third, youth delegations will be asked to nominate youth to participate in the workshop.

### Developing the concept maps

Using the findings from the participatory workshops as a basis for reflection, participants will be asked to participate in a concept mapping study to collectively develop a set of strategies to improve the integration of mental health services within YCs.

Concept mapping enables groups of actors to visualise their ideas about an issue of mutual interest and to develop common frameworks through a structured, participatory process. Qualitative and quantitative data are generated and integrated by participants through sequential steps, beginning with the generation of ideas (brainstorming), structuring of ideas through sorting and rating, development of conceptual maps based on multivariate statistical methods and collective interpretation of the maps.[31] Data analysis will adopt concept mapping techniques with a mixed methodology combining multivariate statistical methods and collective qualitative interpretation.

The research team will develop a focus question to guide the brainstorming activity and with clear instructions to all workshop participants and other relevant stakeholders and young people. Once the refined list of actions is developed, it will be sent again to the participants for sorting and rating. Three different sets of maps (one for each group) will be developed. The data will be analysed using concept mapping techniques, which facilitate visualisation of thematic clusters and identification of areas of consensus for action. The Concept System-CS Global Max software will be used throughout the process.

### Bringing everything together and proposing actions

Finally, the concept maps and all the qualitative and quantitative results will be synthesised to answer our main research question: 'how can the integration of mental health services into YCs in the Swedish health system be strengthened?'. Building on the concept maps and synthesised results, a final workshop will be held with young people and relevant stakeholders to discuss the maps and come up with a set of elaborated strategies.

### Ethics and dissemination

This study has received approval from the Swedish Ethical Review Authority (Dnr 2019-02910, 2020-04720) and complies with the Declaration of Helsinki. All participants will be informed that participation in the study is voluntary. Individuals will be provided with information in verbal and written format, and offered the opportunity to ask any questions. Written informed consent will be obtained from all participants.

Results will be published in open-access peer-reviewed journals and presented at scientific conferences, and as described before, preliminary findings will be discussed with stakeholders during the concept mapping.

## DISCUSSION

This paper presents a study protocol using a multiple case study design to understand the integration of mental health

services within YCs with a view to strengthening FLMHS for youth in Sweden and to develop strategies for strengthening this process.

This study is timely as it goes hand in hand in informing policy efforts to integrate mental health services within YCs and YCs within the FLMHS for youth in Sweden. Atun *et al*'s[28] framework has been used to analyse the integration of programmes on communicable diseases, cardiovascular disease and sexual and reproductive health interventions, but, to the best of our knowledge, it has never been used to analyse the integration of mental health services in primary care. The methodological approach, involving realist evaluation and concept mapping, combines quantitative and qualitative methods sequentially in a way that fits the research questions and the health systems approach to complexity, and is innovative in the field of public health.

The study combines two methodological approaches: realist evaluation[30] and concept mapping.[31] Unlike randomised controlled trials, realist evaluation does not aim to control contextual factors but rather to explore how outcomes of an intervention interact with the context in which it is implemented. Realist evaluation is a methodology that rests on critical realism, and it mainly focuses on individual mechanisms in understanding an intervention. In this study, we do not focus only at the individual level but also the team and organisational levels. This challenges the method to some degree, but we have good experiences of using the method in this broader sense, for example, in the context of primary healthcare services.[38]

Concept mapping is a mixed participatory method for involving stakeholders and youth in a structured conceptualisation process. It has been developed to collect and integrate input from multiple sources and stakeholders with various demand, expertise or interest.[39] One challenge in using concept mapping is that the methodology uses group processes, for example, in brainstorming activities. Interaction within groups and the group processes are affected by entities such as power and status.[40] As participants are diverse in terms of age, education and gender, we see a potential risk that participants will not have equal possibilities to contribute in the concept mapping. To balance participant contributions in group-oriented tasks, we will have this challenge in mind as we put the groups together. The moderator of the session will also be instructed to facilitate the participation of all group members. By taking these measures, we hope to balance the potential negative impact of the group processes to some degree.

The results of the study are expected to contribute new knowledge concerning the process of integration of YC in the FLMHS for youth but also to lead to recommendations for policy-makers and decision-makers within the field of youth mental health services.

## COVID-19 pandemic: adjustments

Due to the COVID-19 pandemic, some adjustments to the planned methodology have been and will be necessary. During year 2020 and 2021, meetings and travels have been restricted in Sweden, with the purpose of minimising spread of the COVID-19 infection. The restrictions have had implications for data collection, all interviews during the pandemic have been and will be performed digitally. The pandemic has also brought an increased strain on the health system in all levels, affecting the possibility to recruit participants operating within the health system, for example, in government authorities or in YCs. This has and will be handled by adjusting the timeline forward and being flexible in terms of how interviews and workshops will be carried out (ie, digital or face to face).

**Contributors** IG designed the research proposal, drafted the application and led the discussion in the group as principal investigator of the research proposal. LRS led the development of the first draft of the manuscript based on the research proposal and gathered comments from all team members to revise successive drafts and develop and submit the final draft of the manuscript to the journal. IG, MC, MW and A-KH are part of the research team that supported and developed the protocol; contributed with ideas and discussed the protocol that was submitted and granted funding from FORTE; and read and critically commented on and revised successive drafts of the manuscript and approved the final version before submission to the journal.

**Funding** This work was supported by the Swedish Research Council for Working Life and Living Conditions (FORTE; grant number: 2018-00364).

**Competing interests** None declared.

**Patient and public involvement** Patients and/or the public were involved in the design, or conduct, or reporting, or dissemination plans of this research. Refer to the Methods section for further details.

**Patient consent for publication** Not applicable.

**Provenance and peer review** Not commissioned; externally peer reviewed.

**ORCID iD**
Linda Richter Sundberg http://orcid.org/0000-0002-5517-0803

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
