## [Reviewer comments · BMJ Open]

ARTICLE DETAILS

TITLE (PROVISIONAL)	How can we strengthen mental health services in Swedish youth clinics? A health policy and systems study protocol
AUTHORS	Richter Sundberg, Linda; Christianson, Monica; Wiklund, Maria; Hurtig, Anna-Karin; Goicolea, I

VERSION 1 – REVIEW

REVIEWER	Helena Tuomainen University of Warwick, Warwick Medical School
REVIEW RETURNED	15-Mar-2021

GENERAL COMMENTS	The article is a protocol of a study designed to evaluate a) the role of youth clinics in the mental health care provision for youth under FLMHS in Sweden and b) how mental health service provision in youth clinics can be strengthened. The introduction provides an overview of the Swedish health system in relation to youth mental health care and clarifies the conceptual framework in relation to the study. The article is very well written, and I have no major comments regarding the study design or methodology. The figures help clarify the design and concepts. However, a third figure depicting the various stakeholders in the FLMHS for youth - the integration of primary care with mental health care - would be helpful. Through the provision of special youth clinics, the Swedish health system is more complex than some other health systems. By providing this additional figure it would make it easier to compare health systems, and to see how the YCs fit in within the larger system in relation to FLMHS. It would also be useful to provide a sentence about the usual care provided by youth clinics. (Line 95-) The title could be improved by cutting one superfluous 'youth': How can we strengthen mental health services in Swedish youth clinics? A health policy and systems study protocol Alternatively: How can we strengthen mental health services for young people in Swedish youth clinics? A health policy and systems study protocol The introduction could be improved by providing a more generic introduction to integrated care options before setting the scene in the Swedish context. This would also limit repetition, currently at the beginning of third and fourth paragraphs. The paragraph about YC's could be added to the previous paragraph, and the first sentence changed: More than 250 youth clinics have been in operation in Sweden since the 1970-1980s, providing youth-centered health services for young people.
---

	Sentence starting on line 99 is too lengthy or complex. Suggest : In this paper, we present the protocol for a health policy and system study with an aim to analyse Swedish policy efforts to strengthen FLMHS for youth. We focus on YCs and adopt Atun's et al conceptual framework for investigating the integration of health interventions into health systems [21]. Line 103: The first research question is ambiguous. Suggest: What is the role and function of YCs in the First Line Mental Health Services for youth in the Swedish health system? Discussion: Additional information on any amendments made to methodology or data collection methods due to Covid-19 would be beneficial. Minor comments & edits Line 27: youth-clinics' Line 35: cases – please add information about what you mean with cases Line 36: including via visual methodologies ... Line 48: the barriers to youth accessing high quality mental health services Line 61: as they have = unlike and suicide rates have not decreased in younger age groups unlike in the overall population Line 62: challenges for Swedish health policies, but is not Line 73-75: Sentence is too complex: the Swedish health system embarked in 2009 to develop and gradually integrate youth mental health services within primary care services, the so-called First-Line Mental Health Services Line 79: have found it Line 99: system study with an aim to analyse Swedish policy efforts to strengthen Line 111: centred on the service user Line 121: influenced by, and influences, both policies and research. Figure 1: Youth clinics (plural) Line 139: advantage, and are less complex and risky Line 145: different interests at stake Line 147: young service users Line 178: have been informed Line 179: previous studies exploring youth views on YC. Furthermore, Line 185: second phase. Stakeholders Line 212: once an intervention has ended or come to a conclusion Line 235: how the intervention is supposed to work Line 265: Specialist mental health professionals will include those working in child mental health services and adult mental health services Line 269-70: (...) we will engage young people in an auto-driven photo-elicitation interview to gather their perspectives Line 275: aged 16-25 (compare line 269) Line 286: This analysis will inform the intervention implementation Lines 294-7: Sentence too complex. (...) participatory process; professionals and young people will be invited to take part in workshops followed by a concept mapping Line 307: YCs involved in the case studies will be asked to invite Line 368: stakeholders with various demands
--	--

REVIEWER	Elizabeth Barrett University College Dublin School of Medicine and Medical Science, School of Medicine
REVIEW RETURNED	14-Apr-2021

GENERAL COMMENTS	Perhaps the introduction could be clearer- what is happening now and what is proposed- for those from outside Sweden. For me, the study questions are clear but come rather later- How and under what circumstances are YCs part of the FLMHS for 104 youth in the Swedish health system? (2) How can the integration of mental health services 105 into YCs in the Swedish health system be strengthened? Really enjoyed the explanation of complexity theory- may be familiar to readers as wicked problems. This is a hugely complex piece of work, as brought home by Table one- each component of this is clearly a study in its own right. You have done well to summarize the detail and breadth of what you are attempting in this ambitious outline. A number of PPI oriented and innovative approaches e.g. photoelicitation are incorporated. Each component presumably has been approved by the ethics committee referred to? How analysis will be completed is not really outlined in detail; likely as this is a description certainly in phase one of several studies contributing to the whole. However I think this would strengthen the paper e.g. perhaps a table in line with Table one, with approaches you plan to take in terms of each component. This is a hugely ambitious project with broad scope. The writing throughout is excellent and engaging.
---

VERSION 1 – AUTHOR RESPONSE

REVIEWER 1		
The figures help clarify the design and concepts. However, a third figure depicting the various stakeholders in the FLMHS for youth - the integration of primary care with mental health care - would be helpful. Through the provision of special youth clinics, the Swedish health system is more complex than some other health systems. By providing this additional figure it would make it easier to compare health systems, and to see how the YCs fit in within the larger system in relation to FLMHS.	Thank you for this good suggestion. We have added a figure that illustrate the stakeholders in FLMHS for youth.	Two clarifying sentences are added and inserted (line 125-128). A new figure 1 is inserted (Line 130)

It would also be useful to provide a sentence about the usual care provided by youth clinics. (Line 95-)	Yes we agree, more information on the usual care provided by YC will clarify.	This is now added (Line 119-125).
The title could be improved by cutting one superfluous 'youth': How can we strengthen mental health services in Swedish youth clinics? A health policy and systems study protocol Alternatively: How can we strengthen mental health services for young people in Swedish youth clinics? A health policy and systems study protocol	We welcome this suggestion and have adopted the first alternative!	Title changed accordingly (Line 1-2).
The introduction could be improved by providing a more generic introduction to integrated care options before	Thank you for bringing this to our attention. We agree that a more generic start will improve the introduction.	We have included a broader paragraph in the beginning of the introduction (Line 65-68)
setting the scene in the Swedish context. This would also limit repetition, currently at the beginning of third and fourth paragraphs.		We have also moved other more universal parts of the introduction to this initial paragraph (Line 68-75) After this, the Swedish setting is introduced and overlapping parts have been removed. (Line 77-95 and following)
The paragraph about YC's could be added to the previous paragraph, and the first sentence changed: More than 250 youth clinics have been in operation in Sweden since the 1970-1980s, providing youth- centered health services for young people.	We agree! Thank you for the suggestion.	The changes suggested are carried through. (Line 114-117)
Sentence starting on line 99 is too lengthy or complex. Suggest : In this paper, we present the protocol for a health policy and system study with an aim to analyse Swedish policy efforts to strengthen FLMHS for youth. We focus on YCs and adopt Atun's et al conceptual framework for investigating the integration of health interventions into health systems [21].	Again, a good suggestion that we welcome.	The sentence is changed in line with the reviewers suggestion (Line 132-135)

Line 103: The first research question is ambiguous. Suggest: What is the role and function of YCs in the First Line Mental Health Services for youth in the Swedish health system?	We agree that this research question is ambiguous. We have still decided to keep the current formulation of the first question. The current formulation "How and under what circumstances..." is grounded in the realist evaluation approach which is the methodological base of the first part of the research project. We consider that by keeping the research question as it is now there is more coherence between the research questions and the methodological design.	Not changed
Discussion: Additional information on any amendments made to methodology or data collection methods due to Covid-19 would be beneficial.	A good suggestion! We have added this information in the discussion	We have added a new sub-heading " Covid-19 pandemic: adjustments" in the end of the Discussion section. We provide with additional information on
		the amendments (Line 430-439)
Minor comments & edits Line 27: youth-clinics'	Thank you for all of these minor, but still important edits!	Revised, line 30
Line 35: cases – please add information about what you mean with cases		Revised, line 41-42
Line 36: including via visual methodologies ...		Revised, line 43
Line 48: the barriers to youth accessing high quality mental health services		Revised, line 55-56
Line 61: as they have = unlike and suicide rates have not decreased in younger age groups unlike in the overall population		Revised, line 80.
Line 62: challenges for Swedish health policies, but is not		This sentence is deleted as it overlapped with another sentence in the background.
Line 73-75: Sentence is too complex: the Swedish health system embarked in 2009 to develop and gradually integrate youth mental health services within		Revised, line 91-95.

primary care services, the so-called First-Line Mental Health Services		
Line 79: have found it		Revised line 98
Line 99: system study with an aim to analyse Swedish policy efforts to strengthen		Revised line 132-133
Line 111: centred on the service user		Revised, line 146-147
Line 121: influenced by, and influences, both policies and research.		Revised, line 157-158
Figure 1: Youth clinics (plural)		Revised, now figure 2
Line 139: advantage, and are less complex and risky		Revised, line 176
Line 145: different interests at stake		Revised, line 181
Line 147: young service users		Revised, line 183
Line 178: have been informed		Revised, line 214
Line 179: previous studies exploring youth views on YC. Furthermore,		Revised, line 215
Line 185: second phase. Stakeholders		Revised, line 221
Line 212: once an intervention has ended or come to a conclusion		Revised, line 249-250

Line 235: how the intervention is supposed to work		Revised, line 273
Line 265: Specialist mental health professionals will include those working in child mental health services and adult mental health services		Revised, line 305-306
Line 269-70: (...) we will engage young people in an auto-driven photo-elicitation interview to gather their perspectives		Revised, line 311-312
Line 275: aged 16-25 (compare line 269)		Revised, line 311
Line 286: This analysis will inform the intervention implementation		Revised, line 328
Lines 294-7: Sentence too complex. (...) participatory process; professionals and young people will be invited to take part in workshops followed by a concept mapping		Revised (sentence divided in two), line 338.
Line 307: YCs involved in the case studies will be asked to invite		Revised, line 350
Line 368: stakeholders with various demands		Revised, line 416

REVIEWER 2		
Perhaps the introduction could be clearer- what is happening now and what is proposed- for those from outside Sweden.	Thank you for this comment, we agree and have revised the introduction accordingly.	We have restructured the introduction. It now starts with a more generic description of the setting and moves to the Swedish context both in terms of youth mental health problems, poor accessibility in the health system and the suggested policy. (Line, 65-102)

For me, the study questions are clear but come rather later- How and under what circumstances are YCs part of the FLMHS for 104 youth in the Swedish health system? (2) How can the integration of mental health services into YCs in the Swedish health system be strengthened?	Thank you for this relevant comment. We have discussed this and still decided to keep the research questions as they are placed in the manuscript now. We find it important to give the reader an introduction to the problem, the key concepts and the health system context before the research questions.	Not changed.
You have done well to summarize the detail and breadth of what you are attempting in this ambitious outline. A number of PPI oriented and innovative approaches e.g. photoelicitation are incorporated. Each component presumably has been approved by the ethics committee referred to?	Yes all studies have been approved by the Swedish ethical committee.	-
How analysis will be completed is not really outlined in detail; likely as this is a description certainly in phase one of several studies contributing to the whole. However I think this would strengthen the paper e.g. perhaps a table in line with Table one, with approaches you plan to take in terms of each component.	Thank you for this comment! We agree that the analysis could be more clear and have addressed this.	Table 1 has been expanded with another column that hopefully gives an overview of the analysis planned. (Line 211) The analysis has also been elaborated in parts of the methods sections, e.g. line 261-262 and line 363-365.

VERSION 2 – REVIEW

REVIEWER	Helena Tuomainen University of Warwick, Warwick Medical School
REVIEW RETURNED	02-Aug-2021

GENERAL COMMENTS	I am happy with the revisions and have no further major revision requests. Minor language edits: Line 35: comprised of Line 67 onwards: Suggest: Young people experience the transition between mental health services (e.g. from child to adult mental health services) as poorly coordinated and dramatic. Line 72: dramatically by about 70% Line 76: mental health services Line 95: youth clinics (YC) Line 99: that fall within the area Line 99: spell out SRH
---

	Line 104: YCs have come to play Line 106: and interact with Line 387: that participants will not have Line 398: adjustments to the planned methodology have been Line 399: meetings and travels have been Line 401: The restrictions have had implications: all interviews during the pandemic have been and will be performed digitally Line 402: The pandemic has also brought an ...
--	--

REVIEWER	Elizabeth Barrett University College Dublin School of Medicine and Medical Science, School of Medicine
REVIEW RETURNED	03-Jul-2021

GENERAL COMMENTS	This is a huge project, with many aspects, which you have well recognised, and you attempt to bring the many voices and stakeholders together in a very laudable way. It has the potential to greatly impact service design and delivery both in Sweden and in approaches taken elsewhere. In this draft, I really appreciated the outline of the methodology and description of this for the reader not necessarily working in this arena often. My only comments are very minor, not really regarding the work itself or the content; but rather minor grammatical/ clarifying points to help improve an already excellent draft.  - Unclear meaning for sentence at line 72 p 6 of the proof, "In Sweden, the prevalence of mental health problems among young people has increased dramatically with about 70% during the period"- This could perhaps be " by about 70%" ? - Line 95 p 7 - "Currently, the sectors that are most frequently involved in FLMHS are primary health care centers, school health services and youth clinics"- It would be helpful to know what services are available in the youth clinics specifically. These clinics are not seen necessarily elsewhere (my own country does not have these)- and likely will differ in terms of how constituted. The paper refers to mental health but I am not clear what the provision is e.g. .Are they GP led? Is this primary care or secondary care in some instances, is there psychiatry? Psychology? Is counselling available? Educational psychology? Social work etc? Or do they refer on to mental health services? It seems there are 250 of these, which is very robust as a network, and you are absolutely right in highlighting the potential for primary care, but no doubt may also mean there are very many different configurations! May be worth acknowledging this. Of course, perhaps they are very uniform and this may help. P20 line 399-" Due to the Covid-19 pandemic some adjustments to the planned methodology has been, and will be necessary. During the year 2020 and 2021 meetings and travels has..."- In English, travels and adjustments are plural - so suggest "Due to the Covid-19 pandemic some adjustments to the planned methodology have been, and will be, necessary. During year 2020 and 2021 meetings and travels have been", again" the restrictions" are plural and "have" been in line 400 and following line. Sorry if this seems pedantic but better to have it correct! Reference 4 check lead author name spelling (Mc Gorry?- think this is correct in Reference 6!
---

	Overall an interesting and enjoyable read and hopefully will lead to very good work!
--	--

VERSION 2 – AUTHOR RESPONSE

Reviewer 1		
Line 35: comprised of	Revised accordingly	Line 35
Line 67 onwards: Suggest: Young people experience the transition between mental health services (e.g. from child to adult mental health services) as poorly coordinated and dramatic.	Revised accordingly	Line 67 onwards
Line 72: dramatically by about 70%	Revised accordingly	Line 74
Line 76: mental health services	Revised accordingly	Line 78
Line 95: youth clinics (YC)	Revised accordingly	Line 97
Line 99: that fall within the area	Revised accordingly	Line 111
Line 99: spell out SRH	Revised accordingly	Line 111
Line 104: YCs have come to play	Revised accordingly	Line 116
Line 106: and interact with	Revised accordingly	Line 118
Line 387: that participants will not have	Revised accordingly	Line 400
Line 398: adjustments to the planned methodology have been	Revised accordingly	Line 411
Line 399: meetings and travels have been	Revised accordingly	Line 412
Line 401: The restrictions have had implications: all interviews during the pandemic have been and will be performed digitally	Revised accordingly	Line 414-415
Line 402: The pandemic has also brought an ...	Revised accordingly	Line 415
Reviewer 2		
- Unclear meaning for sentence at line 72 p 6 of the proof, "In Sweden, the prevalence of mental health problems among young people has increased	Revised accordingly	Line 74
dramatically with about 70% during the period"- This could perhaps be " by about 70%" ?		

- Line 95 p 7 - "Currently, the sectors that are most frequently involved in FLMHS are primary health care centers, school health services and youth clinics"- It would be helpful to know what services are available in the youth clinics specifically. These clinics are not seen necessarily elsewhere (my own country does not have these)- and likely will differ in terms of how constituted. The paper refers to mental health but I am not clear what the provision is e.g. .Are they GP led? Is this primary care or secondary care in some instances, is there psychiatry? Psychology? Is counselling available? Educational psychology? Social work etc? Or do they refer on to mental health services? It seems there are 250 of these, which is very robust as a network, and you are absolutely right in highlighting the potential for primary care, but no doubt may also mean there are very many different configurations! May be worth acknowledging this. Of course, perhaps they are very uniform and this may help.	Thank you for this comment. We realized that international readers might need more information about the services provided in the Swedish Youth Clinics.	We have added some more information about the Youth clinics in lines 99-108. We have also added two new references, number 25 and 26. The reference numbers were changed consequently throughout the manuscript and in the reference list (line 501- 508).
P20 line 399-" Due to the Covid-19 pandemic some adjustments to the planned methodology has been, and will be necessary. During the year 2020 and 2021 meetings and travels has..."- In English, travels and adjustments are plural - so suggest "Due to the Covid-19 pandemic some adjustments to the planned methodology have been, and will be, necessary. During year	Thank you for your careful reading! We agree, we want it to be right, thank you for helping us! We have changed in line with your suggestions.	Line 411-419
2020 and 2021 meetings and travels have been", again" the restrictions" are plural and "have" been in line 400 and following line. Sorry if this seems pedantic but better to have it correct!		
Reference 4 check lead author name spelling (Mc Gorry?- think this is correct in Reference 6!	Revised accordingly	Line 446

VERSION 3 – REVIEW

REVIEWER	Elizabeth Barrett University College Dublin School of Medicine and Medical Science, School of Medicine
REVIEW RETURNED	09-Oct-2021
GENERAL COMMENTS	Thanks for the comprehensive list of changes